# Phage Biocontrol of Bacterial Leaf Blight Disease on Welsh Onion Caused by *Xanthomonas axonopodis* pv. *allii*

**DOI:** 10.3390/antibiotics10050517

**Published:** 2021-05-01

**Authors:** Nguyen Thi Thu Nga, Tran Ngoc Tran, Dominique Holtappels, Nguyen Le Kim Ngan, Nguyen Phuoc Hao, Marta Vallino, Doan Thi Kieu Tien, Nguyen Huan Khanh-Pham, Rob Lavigne, Kaeko Kamei, Jeroen Wagemans, Jeffrey B. Jones

**Affiliations:** 1Department of Plant Protection, College of Agriculture, Can Tho University, Can Tho 90000, Vietnam; nttnga@ctu.edu.vn (N.T.T.N.); tranbvtvK21@gmail.com (T.N.T.); nlkngan08@gmail.com (N.L.K.N.); phuchaobvtv@gmail.com (N.P.H.); dtktien@ctu.edu.vn (D.T.K.T.); 2Department of Biosystems, KU Leuven, 3001 Leuven, Belgium; dominique.holtappels@kuleuven.be (D.H.); rob.lavigne@kuleuven.be (R.L.); 3Institute for Sustainable Plant Protection, National Research Council of Italy, 10135 Turin, Italy; marta.vallino@ipsp.cnr.it; 4Department of Biology, College of Natural Science, Can Tho University, Can Tho 90000, Vietnam; pknhuan@ctu.edu.vn; 5Department of Biomolecular Engineering, Kyoto Institute of Technology, Sakyo-ku, Matsugasaki, Kyoto 606-8585, Japan; kame@kit.ac.jp; 6Department of Plant Pathology, University of Florida, Gainesville, FL 32611, USA

**Keywords:** bacterial leaf blight, bacteriophages, Welsh onion, *Xanthomonas axonopodis* pv. *allii*, biocontrol

## Abstract

Bacterial leaf blight, which is caused by *Xanthomonas axonopodis* pv. *allii*, annually causes significant yield losses to Welsh onion in many producing countries, including Vietnam. In this study, we isolated and characterized lytic phages Φ16, Φ17A and Φ31, specific to *X. axonopodis* pv. *allii* and belonging to a new phage species and genus within the *Autographiviridae*, from four provinces in the Mekong Delta of Vietnam. Moreover, we evaluated their efficacy for the biocontrol of leaf blight in greenhouse and field conditions. When applying the three highly related phages individually or as a three-phage cocktail at 10^8^ PFU/mL in greenhouse conditions, our results show that treatment with Φ31 alone provides higher disease prevention than the two other phages or the phage cocktail. Furthermore, we compared phage concentrations from 10^5^ to 10^8^ and showed optimal disease control at 10^7^ and 10^8^ PFU/mL. Finally, under field conditions, both phage Φ31 alone and the phage cocktail treatments suppressed disease symptoms, which was comparable to the chemical bactericide oxolinic acid (Starner). Phage treatment also significantly improved yield, showing the potential of phage as a biocontrol strategy for managing leaf blight in Welsh onion.

## 1. Introduction

Welsh onion (*Allium fistulosum* L.) is an important vegetable crop in Vietnam and other Asian countries, such as China and India, which are by far the largest onion producers in the world. This vegetable is mainly grown for its green leaves, which are widely used in Asian cuisine. Welsh onion, which is intensively cultivated throughout the year, is unfortunately also prone to bacterial infections. Bacterial leaf blight, caused by *Xanthomonas axonopodis* pv. *allii* (Xaa), is distributed worldwide and is one of the most important diseases for several onion species. Epidemic outbreaks have already been reported in several *Allium* spp. [1]. The bacterium infects all stages of the plant. Primary inoculum commonly comes from infected seeds, seedlings and plant debris [2,3]. The bacterium penetrates through the leaf stomata and disease rapidly progresses during periods of high humidity. Consequently, during wet and rainy seasons, the leaves rapidly collapse and the plant eventually dies [4]. Therefore, crop yields are significantly reduced, which has been reported for many growing areas worldwide [1,5,6,7].

To manage this bacterial disease, farmers mainly rely on chemical bactericides such as copper compounds and antibiotics, which can have negative effects on beneficial microbial communities and promote the development of resistant strains. In addition, overuse of antibiotics in agriculture would promote the transmission of antibiotic resistance genes from plant bacterial pathogens to human pathogens [8,9,10]. Moreover, chemical residues gradually build up in the environment and the food chain, which is undesirable [11,12].

Currently, biocontrol with bacteriophages is considered a promising alternative strategy for bacterial disease management [13,14]. The application of phages has been shown to be successful in controlling both soilborne [15,16] and airborne plant diseases [17,18,19]. For instance, Lang et al. [20] showed that biweekly phage applications could reduce disease severity symptoms of bacterial leaf blight on onion equal to or better than weekly applications of copper hydroxide plus mancozeb.

This study aims at controlling bacterial leaf blight on Welsh onion using phage biocontrol in Vietnam. We screened phages isolated from Xaa infected onion leaves and we selected three promising ones based on host range and plaque/halo size. These were then characterized by whole genome sequencing. The selected lytic phages were evaluated for their potential of controlling *X. axonopodis* pv. *allii* both in vitro and in greenhouse and field conditions.

## 2. Results

### 2.1. Isolation of Bacteriophages of Xanthomonas axonopodis pv. allii, Causing Leaf Blight on Welsh Onion in the Mekong Delta in Vietnam

In total, twelve Xaa strains and ten specific phage isolates were isolated from infected leaf samples collected from five provinces in the Mekong Delta of Vietnam: Vinh Long, An Giang, Can Tho, Bac Lieu and Soc Trang (Table 1). All twelve Xaa strains were able to cause disease during an in planta pathogenicity test.

Next, a host range analysis shows that the number of Xaa strains that are susceptible to each of the ten phage isolates ranges from four to twelve and clearly depends on the phage isolate (Table 2). Five phages (Φ6, Φ16, Φ17A, Φ17B and Φ31) expressed a broad host range with at least 9 out of 12 tested Xaa being susceptible to infection. Phage Φ31 even shows activity against all tested Xaa strains.

Of the twelve Xaa strains, XaaBL11 appears to be the most susceptible to phage infection as all tested phages were able to infect this strain. Moreover, this strain is causing severe symptoms in an in planta pathogenicity test. Therefore, this bacterium was selected as an amplification strain for phage propagation and for further experiments both in vitro and in vivo.

For the five phages with the broadest host range (Φ6, Φ16, Φ17A, Φ17B and Φ31), the plaque/halo diameter was monitored for 72 h. After 24 h incubation, phage Φ31 displayed the largest halo diameter (6.5 mm), compared to the other four phages. Φ16, Φ17A and Φ6 produced a halo diameter of 5.9 mm, 5.2 mm and 4.9 mm, respectively, all significantly bigger than the one for Φ17B (3.9 mm) (Table 3).

After incubation for 48 h, two phages, Φ31 and Φ16, showed the largest halo diameters measuring 11.5 mm and 11.2 mm, respectively. The other three phages showed diameters between 5.9 and 7.2 mm. Finally, 72 h after plating, Φ16 and Φ31 still displayed the biggest halo diameters (12.7 mm and 12.3 mm, respectively), followed by Φ17A (10.3 mm) and phage Φ6 and Φ17B (Appendix A
Figure A1). These data show that the halos around the plaques, produced by the different phages in collection, increase over time, hinting at exopolysaccharides (EPS)-degrading properties associated with the phage tail.

### 2.2. Phage Characterization by Transmission Electron Microscopy and Whole-Genome Sequencing

The three phages showing the biggest halo zones (Φ16, Φ17A and Φ31) were selected for further characterization. Based on transmission electron microscopy (TEM), all three phages were podoviruses, characterized by an icosahedral head and a short tail (Figure 1).

Subsequently, the genomes of the phages were sequenced, assembled and annotated (Figure 2). Based on a BLASTn analysis, phages Φ16, Φ17A and Φ31 have an 81% nucleotide similarity to *Xylella* phage Paz [21]. Furthermore, a Viptree proteome analysis shows that the three selected Xaa phages from the current collection can be classified within the family of *Autographiviridae* and are related to the *Pradovirus* genus. However, as there is only 9% nucleotide similarity between Φ16, Φ17A and Φ31 and *Xylella* phage Prado, they represent a new phage species [22] within the same new phage genus as phage Paz.

Based on phage sequencing and the *Autographiviridae* genome architecture, there are no genes suggesting a temperate lifestyle for these phages. Moreover, the phage genomes do not encode known proteins associated with virulence or antibiotic resistance. Based on the amino acid sequence of the large terminase subunit, phages Φ16, Φ17A and Φ31 have short direct terminal repeat (DTR) sequences (250 bp) at the end of the sequence and hence a phage T7-like packaging strategy [23]. It is therefore unlikely that these phages will be able to transduce genetic material due to packaging artifacts. In conclusion, these phages are suitable candidates for phage biocontrol.

The genomic differences between Φ16, Φ17A and Φ31 are minimal. A variant calling between Φ16 and Φ17A shows a point mutation in Φ17A (A^2858^ to C^2858^ in 96% of the sequencing reads, with an e-value of 1.36 × 10^−114^), which results in a loss of the start codon of Φ16 gp12. In the case of Φ31, a hypervariable region was discovered while mapping the reads of Φ31 on the reference genome of Φ16. This region is located between gp39 (gene encoding the head-to-tail connector protein) and gp40 (gene encoding the scaffold protein). Here, a drop in the read pileup suggests a deletion event of the intergenic region between the two genes. The remaining reads show an accumulation of mutations in this specific intergenic region (Appendix A
Figure A2).

### 2.3. Efficacy of Phages against Bacterial Leaf Blight on Welsh Onion in Greenhouse Conditions

The three selected lytic Xaa phages were evaluated for their biocontrol potential in a greenhouse experiment. Table 4 shows the percentage of infected leaf tissue for the following treatments of Welsh onion plants, which were all infected with Xaa: a monophage treatment, a phage cocktail treatment, no treatment and a commercial bactericide treatment. Nine days after inoculation (dai), a significant difference can be distinguished between each treatment, compared to the non-treated, infected control (67.5% infected leaf area). Similar reductions were obtained for phages Φ16 and Φ17A and the cocktail of the three phages (between 37.2 and 43.3% infected leaf area), which is still significantly higher compared to the diseased area in the Φ31 monophage treated plants (26.6% infected leaf area). The performance of the phages compared to the chemical bactericide was lower, as the area of diseased tissue was 18.3% when the plants were treated with oxolonic acid (Starner).

Disease protection attributed to monophage treatment was the lowest for phage Φ17A (39.8% infected leaf area versus 67.5% for the untreated object) and the highest for phage Φ31 (26.6% infected tissue) at nine days after inoculation. In addition, the phage cocktail protected less compared to the individual phages (43.3% infected tissue). These results correspond to the phage concentrations on the leaf surface (Table 5), since the Φ31 titer on the leaf surface was significantly higher compared to the other individual phages and the phage cocktail 72 h after inoculation. Representative plants from the various treatments in the greenhouse trial are visually shown in Figure 3.

The optimal phage Φ31 concentration needed to treat the Welsh onion plants to achieve maximum disease control was determined by testing phage concentrations ranging from 10^5^ to 10^8^ pfu/mL (Table 6). A concentration as low as 10^5^ pfu/mL significantly reduced the percentage of symptomatic leaf tissue compared to the control. However, phage applications at 10^7^–10^8^ pfu/mL appeared to be optimal as there was an equal, yet lower percentage of infected leaf area measured compared to a treatment with 10^5^ or 10^6^ pfu/mL. Appendix A
Figure A3 provides an indication of the symptoms visible on the onion plants.

### 2.4. Efficacy of Bacteriophage in Controlling Bacterial Leaf Blight in Field Conditions

Phage Φ31 and the phage cocktail were tested for their efficacy in a field trial where the plants were artificially inoculated (Table 7). The disease developed rapidly in the different test plants (four days after inoculation). At three different time points, all treatments, i.e., phage Φ31, three-phage cocktail at titer 10^8^ pfu/mL and Starner, had a disease index and area under disease progress curve (AUDPC) significantly lower than the control. At four days after inoculation, phage Φ31, the phage cocktail and Starner displayed equal disease protection. Nine days after inoculation, Φ31 proved to be significantly better in disease reduction than the phage cocktail with a disease index of 13.1% and 17.8%, respectively. However, 15 days after inoculation, both phage treatments performed equally in disease reduction.

The plants treated with the bactericide and Φ31 still showed higher disease control based on AUDPC at the end of the experiment, with a disease index significantly lower compared to the phage cocktail (Figure 4). This shows that the phage treatment performs equally well as a commercially available bactericide treatment. With regard to actual yield, only plants treated with Starner had a significantly higher yield. However, with regard to the commercial yield levels, both the phage Φ31 treatment and Starner provided significantly higher yields than the control.

## 3. Discussion

### 3.1. Isolation of Promising Bacteriophages for Biocontrol of Xanthomonas axonopodis pv. allii

In this study, we isolated ten Xaa-specific phages displaying different host ranges. Our results indicate that phages are commonly present on infected onion leaves, since seven out of twelve samples contained phages. This ratio is higher compared to other systems such as the walnut—*X. arboricola* pv. *juglandis* pathosystem where 26 phages were isolated from 126 samples [24]. The isolated phages showed different host ranges when screened on all the Xaa strains. Good candidates for phage biocontrol preferably do not have a too narrow host range to maximize the chance that they can also lyse unknown Xaa strains in field conditions [25]. Alternatively, phage cocktails can be applied to further increase this host range. Therefore, out of five promising phages with the widest host range, the three phages (Φ16, Φ17A and Φ31) with the largest halo diameter and broadest host range were selected for further investigation. Interestingly, the three phages having the biggest halos around the plaques also have the broadest host range, which suggests a correlation between halo size and host range characteristics and should be investigated further, since it could be useful information in the in vitro selection of promising phages for biological control.

For the different phages in the collection, we observed halos around the plaques, increasing over time. This can indicate an EPS-degrading capacity by the bacteriophage through the presence of a polysaccharide depolymerase on the phage particles [26,27]. Olszak et al. [28] demonstrated that this capacity is related to the efficacy of the phage in reducing the pathogenicity of the bacterial pathogen, which also should be investigated for the selected phages in future research.

Genome sequencing confirmed the three selected phages Φ16, Φ17A and Φ31 are very closely related podoviruses, belonging to the *Autographiviridae* family, and that they have a lytic lifestyle. Strictly lytic phages are preferred for phage therapy and biocontrol to ensure that horizontal gene transfer between bacteria is limited [14,15,29]. Although the phages are almost identical at the genome level, they showed some minor differences in the host range. The observed SNPs responsible for these differences could be matched to a hypervariable region upstream of gp40 (gene encoding the scaffold protein) in Φ31. In general, these scaffolding proteins function as assemblers of the mature viral particle. Literature shows that these viral proteins have also obtained alternative functions such as immune evasion and receptor recognition [30]. As such, we can hypothesize that indeed changes in transcription or translation of this scaffolding protein are related to the differences in observed host range.

### 3.2. Efficacy of Phages against Bacterial Leaf Blight on Welsh Onion in Greenhouse and Field Conditions

Both greenhouse and field trials were performed to test the efficacy of the isolated Xaa phages in practice. These trials show that indeed both a single phage Φ31 and a phage cocktail consisting of three phages (Φ16, Φ17A and Φ31) are capable of reducing the progression of the disease. Yet, a treatment based on a single phage Φ31 showed higher disease protection compared to the phage cocktail in both conditions. This could be related to differences observed in the ability of the different phages in lysing the host bacterium XaaBL11 in vitro and/or to the fact that the Φ31 concentration in the cocktail is only one-third compared to the monophage treatment. Furthermore, the result could also indicate that these three phages compete for the same receptor and outcompete each other when a cell is infected with different viruses at the same time [31,32]. In this case, a cocktail would also not perform as well. Since the phages are almost identical at the genome level, it is indeed plausible that they compete during coinfection and hence reduce access by the most effective phage Φ31.

In addition, high disease protection of phage Φ31 could also be related to higher multiplication rate on the leaf surface, similar to the observation by Balogh et al. [33] in controlling bacterial leaf spot on tomato caused by *Xanthomonas perforans*. In our experiments, the plants were artificially infected with only one strain of Xaa, which could lead to a higher efficacy of disease control by an individual phage than a phage cocktail. Since, under natural field conditions, plants could be infected with different strains of Xaa, this result should be further investigated since bacteria could more rapidly develop resistance when only one phage is applied [34]. Under natural conditions, it could, therefore, still be useful to apply a cocktail of phages, preferentially having significant differences in infection mechanism. In addition, a cocktail could be less affected by environmental factors, since not every phage is equally sensitive to external conditions [35].

In our greenhouse trials, a phage titer of 10^5^ pfu/mL was shown to already lead to disease reduction. However, higher phage titers resulted in higher levels of disease protection, with the optimal phage titer being 10^7^ or 10^8^ pfu/mL. This is in line with previous observations by Balogh et al. [33] that showed disease control between 10^6^ and 10^8^ pfu/mL, while 10^4^ pfu/mL was ineffective in controlling bacterial leaf spot on tomato caused by *Xanthomonas perforans*.

In our field trials, four applications of single phage or phage cocktail at 10^8^ pfu/mL once before pathogen inoculation and three additional applications (3, 8 and 13 days after infection) reduced the disease index between 35.5% and 43.5% for single phage treatments and between 21.6% and 28.4% for the cocktail treatments. A similar leaf blight disease reduction from 26% to 50% was recorded by Lang et al. [20] when a phage cocktail was applied biweekly at 10^8^ pfu/mL.

Both the study by Lang et al. [20] and Jones et al. [13] indicate that the maximum disease reduction on *Allium* spp. lies around 50%, which could be related to pathogen aggressiveness on these plants in high humidity conditions. Therefore, management of this disease by a combination of phage biocontrol with other control methods could be considered, e.g., with plant activators or antagonistic microorganisms as suggested by Lang et al. [20] and Jones et al. [13], to further improve the level of disease control.

## 4. Materials and Methods

### 4.1. Host Bacterium Isolation and Phage Manipulations

Infected leaf blight samples were collected from different provinces in the Mekong Delta and were used for isolation of Xaa. After surface sterilization, the infected leaves were inspected for bacterial oozing under a microscope. Next, one drop of suspension containing bacterial ooze was plated on King’s B agar medium and streaked for individual colonies and then incubated for 48 h at 25 °C. Single colonies were picked up, then tested for their pathogenicity by spraying bacterial suspensions (OD_600nm_ = 0.3; corresponding to 3 × 10^8^ CFU/mL) on Welsh onion and scoring symptom development and used as a representative host for further experiments.

For phage isolation, the infected onion leaves were chopped and crushed using mortar and pestle. The homogenized leaves were mixed with an equal volume of water and subsequently centrifuged at 6000 rpm. The supernatant was transferred and treated with chloroform at a concentration of 3–5% and incubated for 5 min before another round of centrifugation (6000 rpm, 5 min). Phages were visualized by mixing 100 µL of this supernatant with 10 mL of 0.8% King’s B soft agar containing a bacterial suspension (the bacterial strain isolated from the same leaf sample) and pouring it on an agar plate. After 24 h incubation, individual plaques were picked up with a sterile toothpick and streaked on a fresh bacterial lawn with a cotton swab. A single plaque was harvested in water as a phage suspension and stored at 4 °C.

Phages were routinely amplified by cotton swab streaking of this phage stock on fresh soft King’s B agar plates containing host bacterium. After 24 h incubation at 25 °C, water was added for harvesting the phages, after excluding the remaining bacterial cells by centrifugation and chloroform treatment.

The host range of the different phages was tested by spotting 5 µL of phage suspension on a bacterial lawn containing the test strain. In short, these bacterial lawns of each test strain were prepared by adding 100 µL of bacterial suspension at an OD_600nm_ of 0.3 to 10 mL of King’s B soft agar (0.8%). Plates were incubated for 24 h at room temperature. The lysis zone was recorded for all strains to determine the hot range.

Promising phages with broad host ranges were selected for further experiments. Plaque formation for the different phages was compared by plating 5 × 10^2^ pfu/mL in triplicate (one plate per replicate). The diameter of the halos around the plaques was recorded 24, 48 and 72 h after infection (incubation at room temperature in darkness).

### 4.2. Phage Characterization

The virion morphology was determined by TEM analysis. The phage suspension was first allowed to adsorb for 3 min on carbon and formvar-coated copper–palladium grids, which were then rinsed several times with water. Next, the grids were negatively stained with aqueous 0.5% uranyl acetate and the excess fluid was removed with filter paper. Observations and photographs were made with a Philips CM10 transmission electron microscope (TEM) (Eindhoven, The Netherlands), operating at 80 kV. Micrograph films were developed and digitally acquired at high resolution with a D800 Nikon camera. Finally, the images were trimmed and adjusted for brightness and contrast using the Fiji software [36].

Next, to analyze the genome of these phages, 10^10^ pfu/mL phage suspensions were used for DNA phenol-chloroform extraction [37]. A sequencing library was then obtained using the Illumina Nextera flex kit and sequenced on an Illumina MiniSeq. The reads were assembled and annotated with RAST [38] using the PATRIC platform [39]. Phage sequences were compared to homologous phage sequences on NCBI using BLASTn [40]. Protein sequences were manually verified using BLASTp and Artemis [41] was used to polish the genbank files. Genome maps were drawn using EasyFig [42]. The comparison between the phage genomes was performed by mapping the reads on the reference genome using Bowtie2 [43] and variants were called using iVar [44]. The data were visualized using an integrated genome viewer [45].

### 4.3. Evaluation of the Efficacy of Phage Treatment in Greenhouse Conditions

The first greenhouse experiment was used to compare the efficacy of different phage treatments for controlling bacterial leaf blight on Welsh onion. The experiment was a completely randomized design with six treatments (three monophage treatments, one treatment with a cocktail of three phages (containing 1/3rd of each phage), a control treatment without phage application and a treatment with oxolinic acid). Each treatment included five replicates, each in a different pot.

Thirty-day-old Welsh onion plants were used for experiments. The phage suspension (10^8^ PFU/mL) of each treatment was sprayed over the leaves (25 mL/pot). After 2 h, plants were inoculated with a phage-susceptible Xaa strain (XaaBL11, OD_600nm_ of 0.3) on the leaf surface using a hand sprayer (again 25 mL/pot). The pots were covered with plastic bags for 24 h in darkness, at 25 °C and 100% humidity in a growth chamber. After 24 h, the plastic bags were removed, and the plants were grown in greenhouse conditions. The percentage of infected leaf area was recorded at several time points until the control treatment was almost fully infected. In addition, the bacteriophage density on the leaf surface of differently treated plants was determined at 0, 48 and 72 h after pathogen inoculation (three leaves per plant; three plants per treatment).

In the second greenhouse experiment, the optimal phage titer was determined using the same experimental setup. Pots were arranged completely randomly for five treatment conditions (four different phage titers, i.e., 10^5^, 10^6^, 10^7^ and 10^8^ pfu/mL and one control treatment without phage application).

### 4.4. Evaluation of the Efficacy of Phage Biocontrol in Field Conditions

A field trial was conducted in a 500 m^2^ Welsh onion field in the An Giang province. This experiment was set up as a completely randomized block design, in which four treatments were evaluated: (1) control treatment without the application of phage or chemicals, (2) phage Φ31, (3) a cocktail of three phages (Φ16, Φ17 and Φ31) and (4) bactericide (oxolinic acid) treatment. There were four replicates per treatment and the treatments were applied 30 days after planting the onion seedlings.

Phages were applied (phage Φ31 or the three-phage cocktail) by spraying a phage suspension (10^8^ pfu/mL) at 1 L/25 m^2^ one hour before pathogen inoculation and at 3, 8 and 13 days after pathogen inoculation (dai). Bactericide treatment consisted of Starner 20 WP (20 g oxolinic acid/16 L water) at 1 L/25 m^2^, which was first applied when the percentage infection was around 5–10% (3 dai) and which was repeated at 8 and 13 dai. Pathogen inoculation was done by spraying XaaBL11 (OD_600nm_ = 0.15, corresponding to 10^8^ CFU/mL; 1 L/25 m^2^) on the leaf surface 30 days after planting. The disease index was recorded at 5, 9 and 15 dai. The actual yield (the whole plant) and commercial yield (without infected leaves) were recorded as well.

### 4.5. Statistical Analyses

All statistical analyses were carried out in MSTAT-C (Statistical software developed by the Crop and Soil Science Department of Michigan State University, USA). First, a Shapiro–Wilk test was used to test for normality of the experimental data with or without transformation by taking the square root of each datapoint. Next, Bartlett’s test was run to determine the equality of variances. Finally, means were separated pairwise using Duncan’s or Tukey’s range test, resulting in a significance level letter report.

## 5. Conclusions

During this research, ten bacteriophages were isolated from twelve bacterial blight-infected onion leaf samples and three promising phages Φ16, Φ17A and Φ31 were selected based on their host range and plaque/halo diameter. The three podoviruses are lytic phages based on whole-genome sequencing and form a new phage species. Phage Φ31 shows higher disease reduction compared to phage Φ16, Φ17A and phage cocktail in greenhouse conditions, and the optimal phage titer for disease control lies at 10^7^ and 10^8^ pfu/mL as these concentrations performed equally well. During field trials, phage Φ31 reduced disease symptoms equally compared to the bactericide Starner and provided a significant increase in crop yield.

## Figures and Tables

**Figure 1 antibiotics-10-00517-f001:**
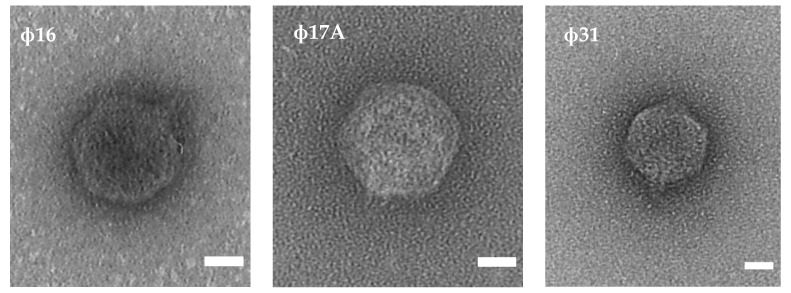
TEM analysis shows the morphology of the three selected phages. All three phage virions contain an icosahedral head and a short tail, typically for the podoviruses. The white scale bar represents 20 nm.

**Figure 2 antibiotics-10-00517-f002:**
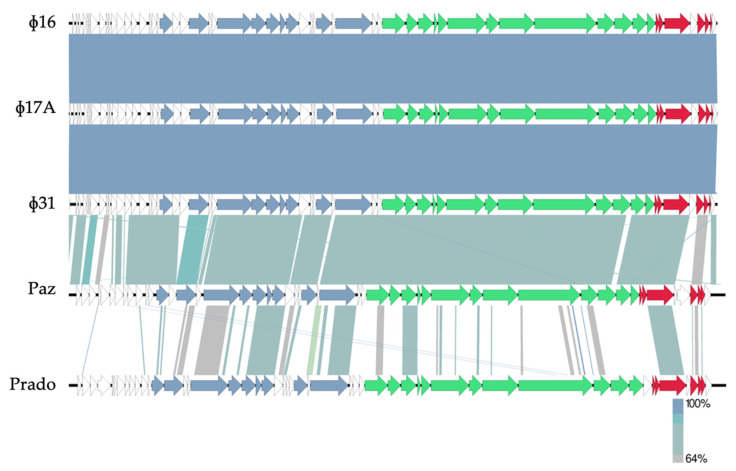
Genome maps of Φ16, Φ17A and Φ31 and comparison between the genomes and related phages Paz and Prado using a BLASTn analysis. The selected phages follow a modular genome organization, typical for lytic phages of the *Autographiviridae*: first an early transcribed region with a lot of ORFan genes, followed by the DNA metabolism region, the structural protein region and the lysis cassette. Arrows represent the different coding sequences: in white—encoding hypothetical proteins, blue—encoding DNA-associated proteins, green—encoding structural proteins and red—encoding the lysis cassette (adapted from EasyFig).

**Figure 3 antibiotics-10-00517-f003:**
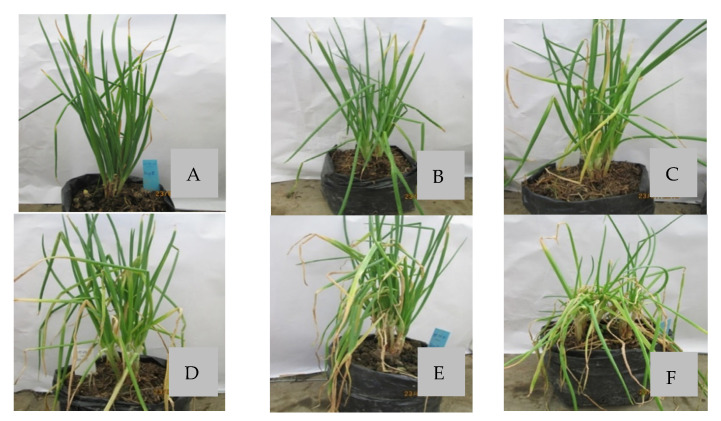
Development of leaf blight caused by X. axonopodis pv. allii on Welsh onion in greenhouse conditions. Five different treatments were tested for their efficacy to control leaf blight symptoms nine days after infection: (**A**) Oxolonic acid (Starner), (**B**) Φ 31, (**C**) Φ 16, (**D**) Φ 17A, (**E**) three-phage cocktail (Φ 16, Φ 17A and Φ 31) and (**F**) Control (bacteria only).

**Figure 4 antibiotics-10-00517-f004:**
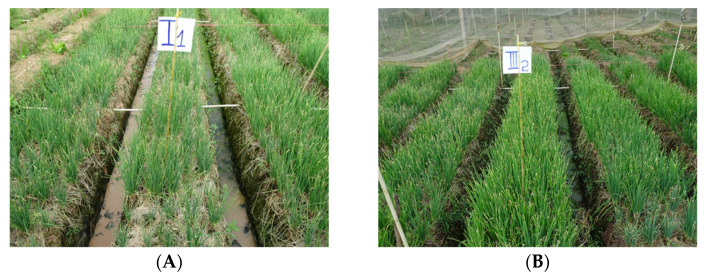
Bacterial leaf blight infection caused by *Xanthomonas axonopodis* pv. *allii* in field conditions. Two different treatments are being shown: (**A**) bacteria only treatment and (**B**) bacteria and phage Φ31 treatment.

**Table 1 antibiotics-10-00517-t001:** Different *Xanthomonas axonopodis* pv. *allii* strains and phages isolated from infected Welsh onion leaves collected from different provinces in the Mekong Delta of Vietnam.

Location (Mekong Delta, Vietnam)	Host Isolate	Bacteriophage Isolate
Binh Tan-Vinh Long	XaaVL02	
Chau Phu-An Giang	XaaAG04	
Binh Tan-Vinh Long	XaaVL05	Φ5A, Φ5B
Binh Tan-Vinh Long	XaaVL06	Φ6
Binh Tan-Vinh Long	XaaVL07	Φ7A, Φ7B
O Mon-Can Tho	XaaCT10	
Hiep Thanh-Bac Lieu	XaaBL11	
Vinh Trach Đong-Bac Lieu	XaaBL12	
Hiep Thanh-Bac Lieu	XaaBL13	Φ14
Hiep Thanh-Bac Lieu	XaaBL14	Φ16
Vinh Trach Đong-Bac Lieu	XaaBL17	Φ17A, Φ17B
Vinh Chau-Soc Trang	XaaST22	Φ31

**Table 2 antibiotics-10-00517-t002:** Evaluation of the host spectrum of ten Xaa phage isolates against twelve Xaa strains. Susceptibility of the Xaa strain to a phage is indicated with a ‘+’, while resistance is indicated with a ‘−‘. The bottom row indicates the total amount of strains susceptible to a certain phage.

Xaa Strain	Xaa Phage									
	Φ5A	Φ5B	Φ6	Φ7A	Φ7B	Φ14	Φ16	Φ17A	Φ17B	Φ31
XaaVL02	+	+	−	+	+	−	+	+	+	+
XaaAG04	−	−	+	−	−	−	+	+	−	+
XaaVL05	−	−	+	+	+	−	+	+	+	+
XaaVL06	−	−	+	−	−	+	+	+	+	+
XaaVL07	−	−	+	+	+	+	+	+	+	+
XaaCT10	+	+	+	−	−	−	+	+	+	+
XaaBL11	+	+	+	+	+	+	+	+	+	+
XaaBL12	−	−	−	+	+	+	−	+	+	+
XaaBL13	+	+	+	−	−	−	+	+	+	+
XaaBL14	+	+	−	+	+	−	+	−	−	+
XaaBL17	−	−	+	−	−	+	+	+	+	+
XaaST22	−	−	+	+	+	+	+	+	−	+
Total	5	5	9	7	7	6	11	11	9	12

**Table 3 antibiotics-10-00517-t003:** Halo sizes of five phages on a lawn of Xaa strain XaaBL11, 24, 48 and 72 h after plating.

Phage Isolate	Location of Isolation	Halo Diameter (mm) ^1^		
		24 h	48 h	72 h
Φ6	Binh Tan-Vinh Long	4.9 b	5.9 c	8.7 b
Φ16	Vinh Trach Dong-Bac Lieu	5.9 a	11.2 a	12.7 a
Φ17A	Hiep Thanh-Bac Lieu	5.2 b	7.2 b	10.3 b
Φ17B	Hiep Thanh-Bac Lieu	3.9 c	6.0 c	6.6 c
Φ31	Vinh Chau-Soc Trang	6.5 a	11.5 a	12.3 a

^1^ Means followed by a different letter (a–c) in the same column do differ significantly (Tukey’s test; *p*-value < 0.05).

**Table 4 antibiotics-10-00517-t004:** Percentage of leaf area infection caused by *X. axonopodis* pv. *allii* (XaaBL11) after different treatments in greenhouse conditions.

Treatment	Percentage of Infected Leaf Area (%) ^1^		
	5 dai ^2^	7 dai ^2^	9 dai ^2^
Φ16	20.2 b	33.2 b	37.2 b
Φ17A	22.6 b	38.9 b	39.8 b
Φ31	11.4 c	22.4 c	26.6 c
Three-phage cocktail	26.3 ab	40.3 b	43.3 b
Starner	7.7 c	15.7 c	18.3 d
Bacteria only	30.5 a	51.2 a	67.5 a

^1^ Means followed by a different letter (a–c) in the same column do differ significantly (Duncan’s test; *p*-value < 0.05). ^2^ Days after inoculation with XaaBL11.

**Table 5 antibiotics-10-00517-t005:** Bacteriophage density (pfu/g leaf) on Welsh onion phyllosphere at various timepoints after pathogen inoculation.

Treatment	Density of Bacteriophage on Leaf Surface (pfu/g leaf) ^1^		
	0 hai ^2^	48 hai ^2^	72 hai ^2^
Φ16	3.53 b	8.55 a	7.35 bc
Φ17A	4.04 ab	8.42 a	7.89 b
Φ31	3.97 b	8.73 a	9.03 a
Three-phage cocktail	4.15 a	7.60 b	6.97 c

^1^ Means followed by a different letter (a–c) in the same column do differ significantly (Duncan’s test; *p*-value < 0.05). ^2^ Hours after inoculation with XaaBL11.

**Table 6 antibiotics-10-00517-t006:** Percentage of leaf area infection caused by *X. axonopodis* pv. *allii* (XaaBL11) in different phage titer treatments of phage Φ31 in greenhouse conditions.

Treatments	Percentage of Leaf Area Infection (%) ^1^		
	4 dai ^2^	6 dai ^2^	8 dai ^2^
10^5^ pfu/mL	8.2 b	21.4 b	43.1 b
10^6^ pfu/mL	6.1 bc	16.6 bc	34.8 c
10^7^ pfu/mL	3.3 cd	9.4 cd	20.3 d
10^8^ pfu/mL	0.6 d	4.9 d	18.4 d
Control	15.9 a	34.1 a	60.0 a

^1^ Means followed by a different letter (a–d) in the same column do differ significantly (Duncan’s test; *p*-value < 0.05). ^2^ Days after inoculation with XaaBL11.

**Table 7 antibiotics-10-00517-t007:** Efficacy of phage and oxolinic acid treatments on the control of bacterial leaf blight of rice caused by *X. axonopodis* pv. *allii* (XaaBL11) and on yield under field conditions.

Treatments	Disease Index ^1^			AUDPC ^3^	Actual Yield (kg/25 m^2^)	Commercial Yield (kg/25 m^2^)
	4 dai ^2^	9 dai ^2^	15 dai ^2^			
Control	18.3 a	23.2 a	28.2 a	288.8 a	31.4 b	26.3 b
Φ31	11.5 b	13.1 c	18.2 bc	178.8 b	34.0 ab	30.9 a
Three-phage cocktail	13.1 b	17.8 b	22.1 b	216.6 b	33.9 ab	29.0 ab
Starner	13.9 b	14.6 bc	15.9 c	189.2 b	34.8 a	31.3 a
*p*-value ^4^	0.0087	0.0030	0.0013	0.0001	0.0238	0.0027

^1^ Means followed by the same letter ((a–c) in the same column do not differ statistically among themselves by a Tukey’s test (*p* < 0.05). ^2^ Days after inoculation with XaaBL11. ^3^ Area under disease progress curve. ^4^ Probability that there are no differences in treatment means according to an analysis of variance.

## Data Availability

The genome data of this novel phage species were submitted to NCBI Genbank and are available through accession number MT951568.

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
