# Peer review of "Phage Biocontrol of Bacterial Leaf Blight Disease on Welsh Onion Caused by Xanthomonas axonopodis pv. allii"

_antibiotics, 2021, doi:10.3390/antibiotics10050517_

Round 1
Reviewer 1 Report
In this paper, a variety of Xanthomonas axonopodis pv. allii were isolated from welsh onion samples from different areas of Vietnam and some phages were identified. Three kinds of bacteriophages with similar genetic relationship were selected for further study. The results of genome sequencing showed that the three kinds of bacteriophages had similar genetic relationship, and the phage ɸ31 with good control effect was the best. It was further verified in greenhouse and field. In view of the discovery in this paper and therefore, we think that this paper has a good innovation, proving that ɸ31 has a good control effect on Xanthomonas axonopodis pv. allii in welsh onion, so it can be considered for publication after modification.
- Slecting bacterial strains, those sensitive to all bacteriophages may not be representative. Can we consider increasing the most resistant bacteria in vivo experiments, and finally determine the practical application effect of bacteriophages.
- Could the bacterial concentration be converted into CFU (Colony-Forming Units) by making standard curve?
- The concentration of the three phages selected in the greenhouse experiment was 108 PFU/ml, and there was no concentration screening before. However, in the later experiments, the concentration screening of ɸ31 was carried out, and the logical relationship here is not reasonable. It is suggested that the concentration screening should be put ahead and the concentration screening should be carried out for all three phages.
- There was no significant difference between 107 PFU / ml and 108 PFU/ml in the concentration screening test results (table. 6). Why not choose 107 PFU/ml?
- Why cocktail therapy be selected?
- The proportion of each phage in cocktail therapy is 1/3. Is there any scientific basis?
- The density of the three phages on the leaf surface is higher, which should be further studied.
Reviewer 2 Report
The manuscript "Phage Biocontrol of Bacterial Leaf Blight Disease on Welsh 2 Onion Caused by Xanthomonas axonopodis pv. allii " is well organized and clearly described paper with high importance for phage-based plant protection. I have only one doubt - that appendix part should be removed and added as a supplementary materials.
Author Response
Reviewer 2 Comments:
The manuscript "Phage Biocontrol of Bacterial Leaf Blight Disease on Welsh Onion Caused by Xanthomonas axonopodis pv. allii" is well organized and clearly described paper with high importance for phage-based plant protection. I have only one doubt - that appendix part should be removed and added as a supplementary materials.
Author Response:
Thank you for these kind words.
In the mdpi template, it says the appendix is an optional section that can contain details and data supplemental to the main text—for example, figures of which representative data is shown in the main text can be added here if brief, or as Supplementary data.
Therefore, we feel the extra figures can stay in the appendix. This makes it easier for the readers to check them instead of downloading them first as supplementary data.
Reviewer 3 Report
Dear Editor and authors,
Nguyen et al. isolated and identified phages with potential for the biocontrol of Xanthomonas axonopodis pv. Allii. The potential for biocontrol was tested in vitro and under field conditions. The genomes of the three most promising phages were sequenced and identified. The work makes significant contributions to the use of leaf blight disease biocontrol. In general, the manuscript is well written. Below, I make some suggestions to make the text clearer. Concerning statistical analyzes, the authors must describe them in more detail and justify the procedures adopted.
Table 3: The authors often refer to "plaque sizes". On lines 105-108, they mention this : "These data show that the plaques, in fact the halos around the plaques, produced by the different phages in collection, increase over time, hinting at exopolysaccharides (EPS) -degrading properties associated with the phage tail".
I think it would be appropriate to make this explanation first. Plate size, at first, sounds strange. Halo diameter would be more accurate.
Statistics: For a posteriori comparison of the halos' mean diameters (Table 3), the authors chose the less conservative mean test of "Duncan". If they want to demonstrate differences in performance between the phages, the authors should have used a more conservative a posteriori test, especially when means exhibit close values.
L114-L115 – "Based on transmission electron microscopy (TEM), all three phages were podoviruses (Figure 1)". Which characteristics evidenced in electron microscopy indicate they are podoviruses? The information presented in the legend of Figure 1. However, the information should be presented at this point in the text.
L278-279 – It is not necessary to repeat the methodology;
L289-290 – In that case, wouldn't it be an alternative to use a phage cocktail with more significant dissimilarity, even if one of them is less virulent?
L395-397 – Statistical analyzes are not sufficiently described. Was an Anova carried out? Any justification for choosing the Duncan test that is less conservative?
L411-413 – Which statistical analysis was used in this case?
